# Fabrication and Optimization of Essential-Oil-Loaded Nanoemulsion Using Box–Behnken Design against *Staphylococos aureus* and *Staphylococos epidermidis* Isolated from Oral Cavity

**DOI:** 10.3390/pharmaceutics14081640

**Published:** 2022-08-05

**Authors:** Niamat Ullah, Adnan Amin, Rana A. Alamoudi, Sheikh Abdur Rasheed, Ruaa A. Alamoudi, Asif Nawaz, Muhammad Raza, Touseef Nawaz, Saiqa Ishtiaq, Syed Shakil Abbas

**Affiliations:** 1Natural Products Research Lab, Gomal Centre of Pharmaceutical Sciences, Faculty of Pharmacy, Gomal University, Dera Ismail Khan 29050, Pakistan; 2Pediatric Dentistry Department, Faculty of Dentistry, King Abdulaziz University, Jeddah 21589, Saudi Arabia; 3Nano Carriers Research Lab, Gomal Centre of Pharmaceutical Sciences, Faculty of Pharmacy, Gomal University, Dera Ismail Khan 29050, Pakistan; 4Endodontic Department, Faculty of Dentistry, King Abdulaziz University, Jeddah 21589, Saudi Arabia; 5Advance Drug Delivery Lab, Gomal Centre of Pharmaceutical Sciences, Faculty of Pharmacy, Gomal University, Dera Ismail Khan 29050, Pakistan; 6Peshawar Dental College, Ripah University, Peshawar 25160, Pakistan; 7Section of Pharmacognosy, Punjab University College of Pharmacy, University of Punjab, Lahore 05422, Pakistan; 8Gomal Centre of Biochemistry and Biotechnology, Gomal University, Dera Ismail Khan 29050, Pakistan

**Keywords:** nanoemulsion, oral bacteria, quorum sensing, essential oil, Box–Behnken design

## Abstract

Oral bacterial infections are fairly common in patients with diabetes mellitus; however, due to limited treatment options, herbal medicines are considered an alternate solution. This study aimed to formulate a stable essential-oil-loaded nanoemulsion for the treatment of oral bacterial infections. Essential oils from edible sources including coriander, clove, cinnamon and cardamom were extracted by hydrodistillation. The response surface methodology was used to optimize the nanoemulsion formulation by applying the Box–Behnken design. The oil concentration, surfactant concentration and stirring speed were three independent factors, and particle size and polydispersity index were two responses. The particle size, polydispersity index and zeta potential of the optimized formulation were 130 mm, 0.222 and −22.9, respectively. The ATR-FTIR analysis revealed that there was no incompatibility between the active ingredients and the excipients. A significant release profile in active ingredients of nanoemulsion, i.e., 88.75% of the cinnamaldehyde and 89.33% of eugenol, was recorded after 24 h. In the ex vivo goat mucosal permeation study, 71.67% of the cinnamaldehyde permeated and that of the eugenol 70.75% from the nanoemulsion. The optimized formulation of the essential-oil-loaded nanoemulsion showed a 9 mm zone of inhibition against *Staphylococcus aureus* and *Staphylococcus epidermidis*, whereas in anti-quorum sensing analysis, the optimized nanoemulsion formulation showed an 18 mm zone of inhibition. It was concluded that formulated essential-oil-loaded nanoemulsion can be used against *S. epidermidis* and *S. aureus* infections in oral cavity.

## 1. Introduction

Oral infections including gingivitis, dental carries and periodontitis are common in diabetic patients and are considered a major reason for tooth loss and severe complications in long-term pervasiveness [1]. Oral bacteria occur either as planktonic cells or present as biofilms [2]. Such bacterial biofilm may be comprised of several mixed and complex microbial communities (more than 700 species of oral bacteria have been identified), and careful management of the biofilm is significant in the prevention of the gradual progression of oral infections [3]. It is well documented that *Staphylococci* are important entities of oral flora, yet debate among researchers continues regarding their role in oral diseases [4]. Despite transient membership of *staphylococci* in the oral cavity, the prevalence of *Staphylococcus aureus* and *Staphylococos epidermidis* varies from 24% to 84% in healthy adult dentate oral cavities or periodontal disease [5,6,7], and an occurrence of 48% has been reported in denture-wearing patients [8], especially in elderly [9].

Treatment of oral bacterial infections is a major challenge in dentistry due to the complex nature of bacterial colonies in the oral cavity [10]. Antibiotics, antiamoebic drugs and quaternary ammonium compounds are generally used for the treatment of oral infections; however, induction of selective pressure in this regard may lead to resistance and toxicity [11]. Thus, there exists a great potential for alternative therapies that are safe, effective and easy to use, such as essential oils.

Essential oils (EO) are lipophilic and volatile liquid extracts that are obtained from plants [12,13]. Essential oils have been used since Ancient Egypt when plant parts were steeped in animal fats and vegetable oils [14]. Essential oils are intended to treat a variety of health concerns, such as cancer, pain, stress, bacterial infections and, more importantly, in aromatherapy [15]. Essential oils are considered potent antibacterial agents [16]; however, an exact mechanism of action of essential oils is not yet known. It has been proposed that interaction with genetic material, interference with enzymatic pathways or interaction with the phospholipid bilayer may have a significant role in the antimicrobial action of essential oils [17].

Nanoemulsions are considered important for enhancing drug delivery due to nanometer range particle size (50–1000 nm) [18], large surface area and enhanced stability, optical transparency, controlled release and flow properties [19]. Several EO-based nanoemulsions have been formulated by researchers that have shown enhanced antimicrobial and antibiofilm potential [20,21]. Based on the importance of EO and nanoemulsions, we designed this project to design and formulate EO-based nanoemulsion for management of oral bacterial infections.

## 2. Materials and Methods

### 2.1. Chemicals and Bacterial Strains

Olive oil (extra virgins) was purchased from the local market, whereas Span 80 and Tween 80 were purchased from Sigma Aldrich USA. The bacterial growth media used included Tryptic Soya Broth (Hi Media, Mumbai, India), nutrient agar (Hi Media, Mumbai, India) and Luria-Bertani Broth (LB) (Oxoid, Hampshire, UK). The standard compounds were purchased commercially, including eugenol (Fluka, Riedstr, Germany) and cinnamaldehyde (Sigma Aldrich, St. Louis, MO, USA).

### 2.2. Essential Oil Extraction and Clinical Strains Isolation Identification

Essential oil (cinnamon and clove) was extracted in lab using hydrodistillation method. Detailed component analysis information was recorded (Appendix A). The bacteria isolated from dental plaques were identified as *Staphylococcus epidermidis* and *Staphylococcus aureus* (Specimen deposited in Pakistan culture bank) using 16S rRNA as reported earlier by our preliminary investigation [22]. The *Chromobacterium violaceum* (DSM 30191) was purchased from DSMZ, Germany.

### 2.3. Preparation of Essential Loaded Nanoemulsion

Nanoemulsion was prepared by the high-speed homogenization method with a slight modification [23]. The oil phase consisted of essential oils, olive oil (oily phase) and span 80, whereas the aqueous phase was comprised of water and Tween 80. After preparing the two phases, the oily phase was added to the aqueous phase dropwise while stirring on a magnetic stirrer. After stirring for 30 min, the mixture was subjected to high-speed homogenization for 10 min.

### 2.4. FTIR Analysis

The compatibility of the active components with any excipients in the nanoemulsion formulation was investigated via FTIR analysis. FTIR analysis of olive oil, the dispersed phase of the nanoemulsion, cinnamon essential oil and clove essential oil, the active ingredients of the nanoemulsion and unloaded nanoemulsion formulation were scanned in the region of 4000–400 cm^−1^ to obtain the IR spectra [24].

### 2.5. Optimization of Formulation by Response Surface Methodology (RSM)

The optimization of the nanoemulsion was performed using response surface methodology (RSM) through Design Expert Software version 11.0.5.0. The effect of three independent variables, including oil concentration (X1), stirring speed (X2), and surfactants concentration (X3), on two responses, particle size (Y1) and Polydispersity Index (Y2), was investigated (Table 1). Three independent variables, oily phase, surfactants concentration and stirring speed with three levels, were used in the Box–Behnken method of optimization (Table 2). A total of 15 randomized runs with three independent variables were carried out, and the findings allowed the best concentrations of the independent variables to be determined to get an optimal formulation (Table 3) [25].

### 2.6. Characterization of Nanoemulsion

#### 2.6.1. Particle Size, Zeta Potential and Polydispersity Index of Nanoemulsion

Purified water was used to make a 100-fold dilution of the emulsion sample. In order to measure the PDI, zeta potential and particle, the 1 mL of the prepared nanoemulsion was diluted 100 times and was injected into a disposable zeta cell (DT1060C) and subsequently into a dynamic light scattering instrument’s measurement chamber (Malvern Zetasizer Nano ZS, UK). After 2 min of equilibration at 25 °C, samples were measured [26].

#### 2.6.2. Heating Cooling Cycle

The prepared optimized formulation was characterized for the cooling–heating cycle to check the thermodynamic stability of the emulsion. The sample was stored at 4 °C for 48 h and then kept at 48 °C for 48 h, and the heating–cooling cycle was repeated three times [27].

#### 2.6.3. Freeze-Thaw Cycle

The optimized nanoemulsion was frozen at −20 °C for 24 h, then thawed at room temperature, and this cycle was repeated thrice, and then it was subjected to the centrifugation analysis to check phase separation [28].

#### 2.6.4. Stability Index

Essential oil nanoemulsion was subjected to three consecutive freeze-thaw cycles, and the stability index of the optimized formulation of nanoemulsion was determined by using the following equation [29].
$$\mathrm{StabilityindexofNE}=\frac{\mathrm{Originalglobulesize}-\mathrm{Changeinglobulesize}}{\mathrm{Originalglobulesize}}\times 100$$

#### 2.6.5. Macroscopic Stability

The prepared formulation stability was determined using centrifugation method [30]. The optimized formulation of the emulsion was subjected to centrifugation analysis to check the kinetic stability. Formulations were centrifuged at 1000, 2000 and 3000 rpm for 15 min. After that, the macroscopic stability of the formulation was determined by comparing the appearance of the emulsion before and after the centrifugation cycle.

#### 2.6.6. pH of the Nanoemulsion

The pH of nanoemulsion was measured with a pH meter that had been calibrated at room temperature (25 ± 2 °C). Prepared nanoemulsion systems were investigated without any dilution [30].

#### 2.6.7. Drug Content

Essential oil contents were determined by using a spectrophotometric method (UV-vis). Nanoemulsion formulation was diluted (1:1000) (nanoemulsion–ethanol) and homogenized in an ultrasound bath and finally quantified at 230 nm for clove and 290 nm for cinnamon essential oil. The essential oil contents were determined from the standard curve already prepared [31].

#### 2.6.8. Encapsulation Efficiency of Nanoemulsion

The nanoemulsion encapsulation efficiency was determined by method with slight modifications [32]. It was calculated by determining the difference between the quantified free essential oil and the initial amount of the essential oil added to the formulation. The following equation was used:EE(%)=[(initialconc:ofEO−freeEO)/initialconcentrationofEO]×100

#### 2.6.9. Release Profile and Drug Release Mechanism

The tests were conducted in Franz diffusion cells (Permegear, model 4G01-00-05-05) with a 7 mL acceptor compartment capacity and a 0.2 cm2 diffusion area. Cellulose membranes (14 kDa, Dialysis tubing cellulose membrane, Sigma-Aldrich, St. Louis, MO, USA) were immersed in receptor fluid for 24 h before being put between the donor and receptor compartments. The trials were conducted at a temperature of 37 °C ± 1 °C. The receptor fluid was phosphate buffer solution at pH 6.8 to simulate physiological circumstances (oral cavity). Samples were taken at regular intervals (0.5, 1, 1.5, 2, 4, 8, 12, 16 and 24 h) and evaluated with a UV spectroscope at 290 nm for Cinnamaldehyde and 230 for eugenol. The receptor compartment was supplied with fresh receptor fluid after each withdrawal. The cumulative amount of essential oil released vs. time was plotted to quantify clove oil and cinnamon oil release across the membrane [33].

#### 2.6.10. Ex Vivo Permeation

Ex vivo permeation study was performed using goat buccal mucosa. The goat buccal mucosa was obtained from a local slaughterhouse instantaneously; as the goat was slaughtered, the buccal mucosa was isolated and kept in phosphate buffer pH 6.8 till further use. The buccal mucosal membrane was separated from the underlying tissues with the help of veterinary doctor. The buccal membrane was placed between the donor and acceptor compartments of the Franz diffusion cell. Then, 0.5 g of the nanoemulsion was placed in the donor compartment, and the Franz diffusion cell was maintained at a temperature of 37 ± 1 °C. The acceptor compartment was filled with phosphate buffer pH 6.8, and sampling was conducted at time regular intervals (0.5, 1, 1.5, 2, 4, 8, 12, 16 and 24 h). Permeated amount of the essential oil was analyzed through a UV spectrophotometer at 290 nm for cinnamon essential oil and 230 nm for clove essential oil [34].

#### 2.6.11. Skin Irritation Test

A skin irritation test was performed with slight modifications [35]. Briefly, Wister rats (200–250 gm) were divided into 2 groups; group 1 was treated with the optimized formulation of essential-oil-loaded nanoemulsion, and group 2 was positive control, and formalin 0.8% *v/v* was used as a positive control while blank formulation was taken as negative control. Rats were observed for any type of skin irritation symptoms such as skin irritation, redness and erythema formation. The results were coded in the form of yes or no after 24, 48 and 72 h. The above-mentioned test was performed to check any sort of irritation on the skin in general if the same formulation would apply to skin and soft tissue infections. However, the skin (buccal mucosa) irritation potential of the optimized formulation was investigated to note compatibility of formulation in oral mucosa of goat oral cavity.

### 2.7. Antimicrobial Activity

Disc diffusion method was used to determine the diameter of the zone of inhibition using Mueller–Hinton. The 24 h old strain of the bacteria was spread on the media. Afterward, test samples were applied on sterilized blank discs already placed on each plate. Plates were incubated in an oven (37 °C) for 24 h, and later inhibition zones were measured [36].

### 2.8. Anti-Quorum Sensing Activity

Anti Qs activity was measured using standard protocol [22]. Briefly, Luria Bertani (LB) agar plates were, prepared and 24 h old strain of the *C. violaceum* (1/100 ratio) was steaked on it. Afterward, test samples were applied on sterilized blank discs already placed on each plate. Plates were incubated in an oven (30 °C) for 24 h, and later inhibition zones were measured. After 24 h zone of inhibition was measured and results were recorded.

### 2.9. Statistical Analysis

The analysis of variance (ANOVA) of individual responses was controlled using Design Expert software version 11.0.5.0.

## 3. Results

### 3.1. ATR-FTIR

ATR-FTIR spectra of optimized nanoemulsion formulation as well as physical mixtures with various excipients revealed a lack of any sort of incompatibility. In the ATR-FTIR spectrum of the olive oil peaks, 2920 cm^−1^ and 2856 cm^−1^ are associated with OH and fatty acid stretching, while 1746 cm^−1^ represents the ester C=O group. In the clove oil spectrum, 3543 cm^−1^ represents the OH stretching, while 1511 cm^−1^ represents the aromatic C=C and phenolic group. In the case of the spectrum of cinnamon oil, 3465 cm^−1^ is associated with OH stretching, 1667 cm^−1^ represents the C=O group and 1619 cm^−1^ represents the C=C group (Figure 1). Now in the spectrum of the loaded nanoemulsion, all the major spectra of the active ingredients are present, which confirms that there is no incompatibility between the active ingredients and the excipients.

### 3.2. Optimization of Essential Oil Nanoemulsion

Response surface methodology was adopted to optimize the essential-oil-loaded nanoemulsion. Box–Behnken design was used to optimize the essential oil nanoemulsion (Table 4). The effect of the independent factors was checked on the dependent variables (response) using a 3D response surface plot and contour plot (Figure 2, Figure 3, Figure 4 and Figure 5). In order to check the individual and combined effect of each independent factor on the dependent factors (responses), Design Expert Software (version 11.0.5.0) was used. The quadratic effect was the best to utilize on all parameters because it has the greatest impact both separately and in combination. The analysis of variance (ANOVA) of individual responses was managed by Design-Expert software, and the outcome indicated model fitting for data sets. (linear, 2FI, quadratic). The independent factors: oil concertation (X1), Stirring speed (X2), and Surfactants concentration (X3) were evaluated at three levels (low −1, medium 0, high +1) to formulate essential-oil-loaded nanoemulsion. Oil (X1) was used in different concentrations such as 10% *w*/*w*, 15% *w*/*w* and 20% *w*/*w*. Stirring speed (X2) was at three different speed 10,000 rpm, 14,000 rpm and 18,000 rpm, while surfactants (X3) was also in three-level that is 1.5, 2.5 and 3.5% *w*/*w*. A total of 15 formulations were prepared and fed into the Design-Expert software. The particle size (Y1) of formulation F5 was 130 nm which was the smallest of all 15 formulations, as shown in Table 4. Similarly, the particle size of formulation F12 was 561.6 nm, the highest in all 15 formulations. Polydispersity Index (PDI) (Y2) of formulation F5 was the lowest and was 0.222, while formulation F12 had the highest PDI of 0.401.

Overall summary of statistical analysis and different models such as Linear, 2FI and Quadratic models were applied, and according to the Design-Expert software using Box–Behnken design, the best fit model was quadratic (Table 5, Table 6 and Table 7).

### 3.3. Effect of Independent Variables on the Particle Size of Essential Oil Nanoemulsion (Y1)

Three-dimensional surface and contour plots were used to investigate the effect of independent factors on particle size (Y1) of the essential-oil-loaded nanoemulsion. It was observed that the oil concentration (X1) has a positive effect on the particle size of the nanoemulsion. Particle size increased from 130 nm to 561.6 with an increase in the oil concentration from 10% to 20%, provided all the other factors were kept constant, while surfactants concentration and stirring have a negative effect on particle size, increasing the concentration of surfactants and stirring speed reduced the particle size from 561.6 to 130 nm.

The model *p*-value in Table 8 is 0.0012, which is less than 0.05, indicating that it is significant. Furthermore, the model’s greatest F-value of 25.26 indicates that it is significant (Table 8). An F-value of this magnitude has a 0.12 percent chance of occurring due to noise.

Model terms with *p*-values less than 0.0500 are significant. X1, X2, X1X2 and X22 are important model terms in this scenario (Table 8)
Particle size(Y1) = +356.57 + 114.92X1 − 85.93X2 − 4.62X3 − 100.82X1X2 + 2.77X1X3 − 8.82 X2X3 − 35.45X12 − 76.15 X22 + 36.25 X3

Independent concentration X1 has a positive effect on particle size in the above-mentioned polynomial equation, whereas X2 and X3 have a negative effect. The best fit of the model was indicated by the lowest *p*-value and the highest F-value (Table 9).

### 3.4. Effect of the Independent Variables on Polydispersity Index (Y2)

Contour and 3D graph show that by keeping the other parameter constant, an increase in oil concentration increased the PDI values of the nanoemulsion from 0.222 to 0.401, which shows that oil concentration has a positive effect on the PDI of the globules of the nanoemulsion, while surfactant concentration and stirring speed have a negative effect on the PDI of the nanoemulsion, increasing these two factors reduced the PDI from 0.401 to 0.222.

The model has an F-value of 40.28 and a *p*-value of 0.0004, indicating that it is significant. Model terms with *p*-values less than 0.0500 are significant. X1, X2 and X32 are significant in this scenario.
PDI(Y2) = +0.2883 + 0.0574X1 − 0.0401X2 − 0.0068X3 − 0.0105X1X2 0.0133X1X3 + 0.0018X2X3 + 0.0076X12 + 0.0051X22 + 0.0238X32

The above quadratic showing that oil concentration (X1) has a positive effect on the PDI of the nanoemulsion means the PDI will increase with an increase in the oil concentration while stirring speed (X2) and surfactants concentration (X3) have a negative effect on the PDI of the nanoemulsion. The software compares various variables besides considering individual variables. Since our *p*-value (<0.05) was considered as the level of significance, all individual variables were significant, whereas when variables were combined by software, the results were insignificant. The uniqueness of Design-Expert is that it compares individual and combined variables variable by itself. The data revealed individual variable significance in the manuscript; we proceeded with individual variables and calculated responses.

### 3.5. Zeta Potential of the Formulation

The zeta potential of all five formulations was determined; the lowest zeta potential was recorded for F4 (−7 mV), whereas the highest was observed for F5 (−22.9 mV) (Table 10). We used F5 for further analysis since the optimum zeta potential is ±25.

### 3.6. pH

The pH values of all the formulations in Table 11 were adjusted up to 7.4 to simulate the pH value of the buccal cavity (Table 11).

### 3.7. Particle Size, Zeta Potential and Polydispersity Index of the Optimized Essential-Oil-Loaded Nanoemulsion

The optimized formulation ((Oil 10% (*w*/*w*), Surfactant 2.5% (*w*/*w*), essential oils 3% (*w*/*w*)) of the prepared nanoemulsion was characterized by particle size, zeta potential, and polydispersity index (Figure 6 and Figure 7). It was observed that the maximum zeta potential was recorded in the case of formulation F5 (−22.9 mV), and formulation F4 had the lowest zeta potential (−7 mV). Likewise, the PDI value of optimized formulation F5 was 0.222, and it was considered moderately dispersed.

### 3.8. Macroscopic Stability

The optimized nanoemulsion formulation was subjected to centrifugation analysis to check the kinetic stability of the prepared nanoemulsion. It was centrifuged at 1000, 2000 and 3000 rpm for 15 min; there was no phase separation, and the formulation was found stable (Table 12).

### 3.9. Heating Cooling Cycle

After passing the centrifugation test, the prepared optimized nanoemulsion was subjected to a heating-cooling cycle to determine the thermodynamic stability. It was observed that after three successive heating-cooling cycles creaming and phase separation of nanoemulsion occur, which confirmed that nanoemulsion is thermodynamically unstable.

### 3.10. Freeze-Thaw Cycle

The optimized nanoemulsion formulation was subjected to a freeze-thaw cycle, and it was observed that the nanoemulsion showed no phase separation after three successive freeze-thaw cycles, but the particle size was increased from 130 nm to 151 nm, and PDI increased from 0.222 to 0.331.

### 3.11. Stability Index of Nanoemulsion

It was determined that the stability index of the optimized formulation of the nanoemulsion was 83.3%.

### 3.12. Drug Contents

Clove essential oil and cinnamon essential oil nanoemulsion (Tween 80 as surfactants and Span 80 as cosurfactants) was produced via a high share homogenization method that exhibited homogeneous drug distribution within the formulation. The optimized nanoemulsion formulation of cinnamon essential oil and clove essential oil showed essential oil contents values of 97.9 ± 1.34% and 95.36 ± 0.45%, respectively. The findings of the drug content experiment confirm that % the drug content was within the USP official limit, which is 100 ± 10%.

### 3.13. Encapsulation Efficiency of the Nanoemulsion

Entrapment efficiency refers to the quantity of drug entrapment within a nano fomulation compared to the initial drug concentration in the formulation. Table 13 shows the average entrapment efficiency of the nanoemulsion.

### 3.14. Drug Release

A released study of the nanoemulsion was performed using the Franz diffusion apparatus. It was observed that after 24 h, cinnamon oil (cinnamaldehyde) was released 88.75% from the nanoemulsion, and clove oil (eugenol) released 89.33% (Figure 8).

### 3.15. Mechanism of Drug Release

Mechanism of the drug release from the nanoemulsion was determined by applying different kinetic models zero order, first order, Higuchi and Korsmeyer–Peppas model (Figure 9 and Figure 10).

The R2 values for each formulation are given in Table 14. The best fit model was Korsmeyer–Peppas, having an R^2^ value of 0.99, which indicated that it was a diffusion-controlled drug release from the prepared formulation.

### 3.16. Ex Vivo Permeation

Goat buccal mucosa was used instead of the cellulose membrane in the ex vivo permeation study. It was found that after 24 h, 71.67% of the cinnamon oil (cinnamaldehyde) was permeated, and 70.75% of the clove oil (eugenol) was permeated through goat buccal mucosa (Figure 11).

### 3.17. Antibacterial Activity of Nanoemulsion

Antibacterial activity of the prepared optimized formulation of nanoemulsion was performed against isolated oral bacterial strains, and significant inhibition of *S. epidermidis* (9 mm) and *S. aureus* (9 mm) was recorded (Table 15). The activity of the blank formulations was also checked to confirm that the observed activity is due to the essential loaded in the nanoemulsion. It was observed that the optimized formulation exhibited a very clear zone of inhibition against both the oral bacterial strains. The blank formulation showed no zone of inhibition because no essential was loaded (Appendix A)

### 3.18. Antiquorum Sensing Activity of Nanoemulsion

Antiquorum sensing of the prepared formulation of nanoemulsion was performed using *Chromobacerium voilaceum*. The loaded nanoemulsion showed significant inhibition (16 mm) compared to unloaded (0 mm) (Table 16; Appendix A).

### 3.19. Skin Irritation Test

A skin irritation test of the optimized formulation of essential-oil-loaded nanoemulsion was performed using formalin as a positive control. The skin irritation results are given in Table 16. The prepared optimized nanoemulsion was found to be safe and nonirritant. This may be due to the encapsulation of essential oils in olive oil.

## 4. Discussion

ATR-FTIR analysis is performed to determine incompatibility between the excipients and active components of the formulation. ATR-FTIR was carried out for olive oil, cinnamon oil and clove oil, representing the oily phase and active ingredient of nanoemulsion formulation. Similarly, ATR-FTIR analysis extended for loaded as well as unloaded nanoemulsion formulation, and it was observed that active ingredients peak with fewer intensities confirmed encapsulation of the essential oil in the nanoemulsion formulation [37].

The particle size of the nanoemulsion is directly proportional to the concentration of the oil (dispersed phase in o/w emulsions). This increase was attributed due to the competition of the oil particles for the emulsifying agent that remains in the emulsifying chamber for a limited amount. Due to the presence of a limited concentration of the emulsifier in nanoemulsions, the smaller particles formed during homogenization started the coalescence process, and particle size increased [38]. The particle size of the nanoemulsion increases with an increase in the dispersed phase [39]. The second factor was stirring speed (X2), and it was noted that stirring speed has a negative effect on the particle size (Y1) of the nanoemulsion that by increasing the stirring speed, the particle size of the nanoemulsion is reduced. As we increased the stirring speed from 10,000 rpm to 18,000 rpm, the particle was reduced from 561 nm to 130 nm. Surfactants/oil phase are evenly distributed in the aqueous phase probably due to proper stirring, and this phenomenon enables the formation of small droplets of the nanoemulsion, and it is clearly mentioned in the literature that the particle size of the nanoemulsion is inversely proportional to the stirring speed [40] as reported earlier [25,41]. The stirring process provides the necessary energy for the emulsion system and greatly affects the stability of the emulsion. Moreover, intense stirring can produce a smaller droplet size in the emulsion [41]. The particle size of the nanoemulsion is reduced due to the fact that high energy creates a deforming force that overcomes the pressure of Laplace and breaks the particle into smaller sizes [42]. The Span 80 and tween 80 are non-ionic surfactants and possess an outstanding stabilizing effect. It was observed that the globule size of the nanoemulsion was reduced with an increase in surfactant concentration. Moreover, previous literature revealed that the particle size of the nanoemulsion decreases with an increase in the concentration of the surfactant [43]. There are two factors involved in the stabilization of the emulsion: (1) the greater the emulsifying agent level, the more stable the oil-water interface area; (2) at a higher concentration of the emulsifying agent, the surface of oil particles is covered completely, and this phenomenon reduces the chances of the instability process the coalescence [44]. Non-ionic surfactants such as span and tween have the tendency to produce smaller particle sizes due to the fact that these types of surfactants can easily adsorb at the surface of the particle [45].

The oil concentration in oil–water nanoemulsion formulation has a positive effect on the PDI. This effect may be due to the fact that when the oil concentration is increased at a fixed surfactant concentration, lesser surfactants molecule are available to coat the increased oil particles’ surfaces, and hence, due to higher interfacial tension and lesser amount of surfactants available, coalescence may occur which ultimately leads to increased PDI [46]. The change of PDI was associated with the particle size, indicating that the large particle aggregate broke into more uniform and small particles under microfluidics treatment [47]. Polydispersity Index (PDI) is one of the parameters which describe the quality of the emulsifying process. Homogeneity of the emulsion depends upon the polydispersity Index, and resistance of the emulsion to creaming depends upon the homogeneity of the particle size. Commonly PDI values less than 0.10 are considered highly monodispersed, values from 0.10 to 0.40 are considered moderately dispersed and values more than 0.40 are considered highly dispersed [48].

Moreover, the effect of the stirring speed was investigated on the polydispersity index (PDI). It was noted that increasing the stirring speed decreases the PDI of the formulation. The polydispersity index indicates the homogeneity in particle size distribution in the pharmaceutical formulations [49]. Stirring speed is inversely proportional to the PDI of the formulation; this may be due to the fact that stirring speed provides deforming energy to the globules of the emulsion and decrease the globule’s size and decrease the PDI of the emulsion [50]. It was observed that keeping the other factors constant, an increased surfactant concentration significantly reduced the PDI of the essential-oil-loaded nanoemulsion. This could be due to the fact that more surfactant molecules are available to coat the oil particles and prevent the process of coalescence, which contributes to higher uniformity in particle size and better stability [51].

The zeta potential of all the 15 formulations was determined using a zeta sizer. It was observed that the maximum zeta potential was obtained by the formulation F5, which was −22.9 mV, and the formulation F4 had the lowest zeta potential, which was −7 mV. Moreover, it was noted that the zeta potential of the formulations increased with a decrease in the particle size of the nanoemulsion. This may be due to the fact that when the particle size decreases, the surface area increases, due to which surface charge also increases. Zeta potential is an analytical technique used to measure the surface charge of NPs in colloidal dispersions. An opposite charged thin layer is attracted by the surface charge of particles and binds to it, forming a thin layer called the stern layer. A stern layer is formed when particles diffuse in a solution, and an outer diffuse layer is formed by loosely associated ions, as a result of which an electrical double layer is formed [52]. The zeta potential is an important parameter for consideration of the short- and long-term stability of emulsions [53]. The intensity of the zeta potential suggests colloidal stability. Low zeta potential values promote coagulation, flocculation and aggregation due to van der Waals forces, while high zeta potential values allow coagulation, flocculation and aggregation [54,55]. Increasing the stirring speed decreases the particle size and increases the surface charge of the particle because as the particle size decreases, the surface area increases, due to which the surface charge also increases [56].

Nanoemulsions are thermodynamically unstable [35]; thus, they are likely to display creaming/phase separation in long-term storage. Nanoemulsions are kinetically stable [57]. Thus, the stability of freshly prepared nanoemulsion was determined by applying stress conditions (centrifugation) to accelerate emulsion breakage. In nanoemulsion, the free energy of the colloidal dispersion is greater than the free energy of the separate phase, which means that the nanoemulsions are thermodynamically unstable and can be made kinetically stable by ensuring that there is a large energy barrier between the two phases [58].

After three successive freeze-thaw cycles, the formulation was stable, but particle size was slightly increased from 130 nm to 151 nm, and PDI increased from 0.222 to 0.331. The increase in droplet size could be explained as there might be crystallization during the freeze-thaw cycle, which causes breakage of the interfacial film of surfactants around the droplets, coalescence of the droplets as well as separation of the two immiscible phases (water and oil). It was noted that the PDI of the nanoemulsion was increased, which could be due to the accumulation of some oil particles in the nanoemulsion [43]. The centrifugal stability was not disturbed, which indicates no phase separation after centrifugation at 3000 rpm for 10 min. Although there was a slight increase in oil particle size and PDI of the nanoemulsion, indicating that the overall formulation of nanoemulsion had an outstanding freeze-thaw cycle.

Higher drug solubility in combination with specific oily phases, as well as drug compatibility with other constituents, contribute to encapsulation efficiency and system homogeneity. The medication’s insoluble nature causes it to become entrapped in an oil globule, which can be stabilized by using surfactants and co-surfactants that have the opposite effect on drug encapsulation. This could be owing to a higher surfactant content, which results in smaller particle sizes and thus reduced drug trapping within nanoemulsions [25]. Furthermore, drug partitioning increased the solubility of active ingredients from the oily to the aqueous phases, resulting in a reduction in formulation viscosity and an improvement in the diffusion phase during self-assembling, demonstrating a conclusive reason for lower active ingredient entrapment efficiency in the formulation.

Chronic gum infection such as periodontitis has a very complicated heterogeneous microbial population consisting of Gram-negative and Gram-positive microorganisms [59]. The microbiological features of caries and periodontal disease are fairly diverse in healthy people compared to diseased patients. In both cases, co-association of different organisms into consortia has been reported [60]. As explained earlier, *staphylococci* are transient members of the oral cavity; however, they have a critical role in infections in denture-wearing patients. Thus, treating such infection is of great importance. Our successfully designed stable nanoemulsion was analyzed for its antimicrobial and anti-qs properties in vitro and it was evident that essential-oil-loaded nanoemulsion showed promising inhibition of clinical isolates that may be due to synergistic effect of individual components, particularly phenolics and terpenes as reported earlier [61]. These findings are considered important since clove and cinnamon oils have traditional usage and are considered safe and economical.

## 5. Conclusions

The efficient essential oil delivery methods before they are included in different dosage forms have drawn more and more attention over the past several decades. The possibility of using essential oils as sources of antimicrobial agents in treatments, food preservation and packaging has received much consideration. Due to problems such as limited solubility, solvent toxicity, volatility and strong organoleptic taste, their commercial application has been constrained. Due to their biocompatibility, biodegradability, nontoxicity and target selectivity, nanoemulsions are a strong contender for the formulation of essential-oil-based antimicrobial nano systems. It was concluded that cinnamon and clove essential-oil-based formulation showed promising antibacterial and antiquorum sensing activity through encapsulation and delayed release from the optimized oil in water nanoemulsion formulation.

## Figures and Tables

**Figure 1 pharmaceutics-14-01640-f001:**
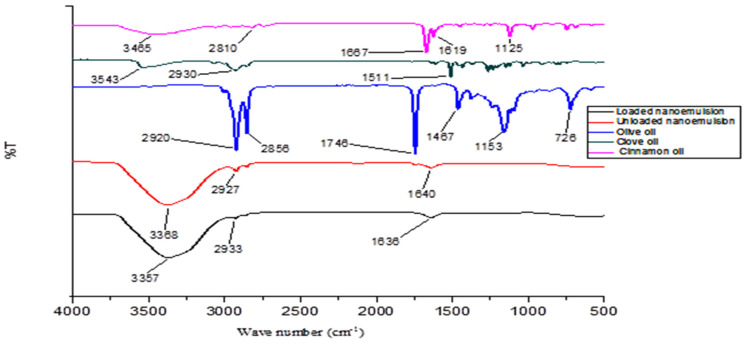
FTIR spectra of olive oil, cinnamon oil, clove oil, essential loaded nanoemulsion and unloaded nanoemulsion.

**Figure 2 pharmaceutics-14-01640-f002:**
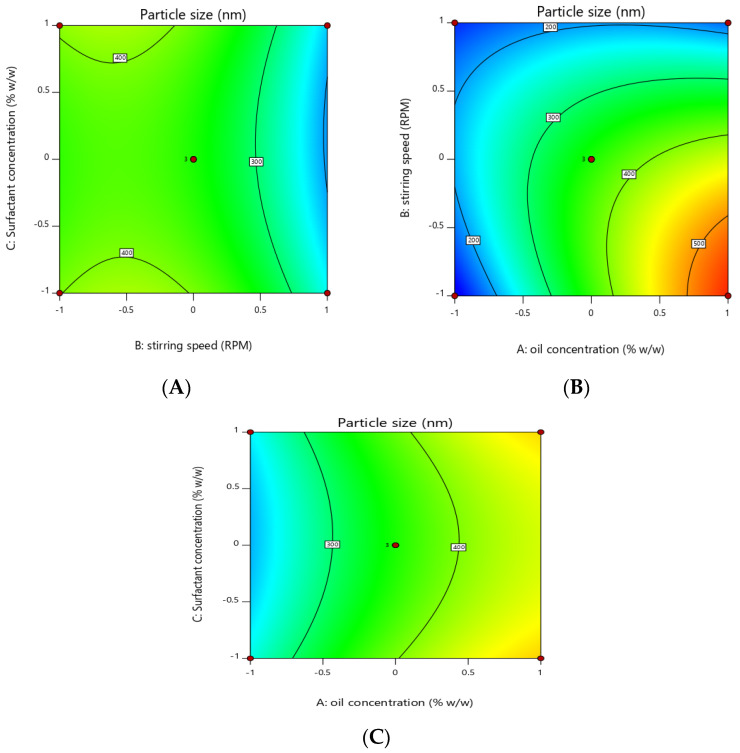
Contour plot showing the effect of independent factors X1 (oil concentration), X2 (stirring speed) and X3 (surfactants concentration) on responses Y1 and Y2. (**A**) effect of surfactant concentration and stirring speed on particle size (**B**) effect of stirring speed and oil concentration on particle size. (**C**) effect of surfactant concentration and oil concentration on particle size.

**Figure 3 pharmaceutics-14-01640-f003:**
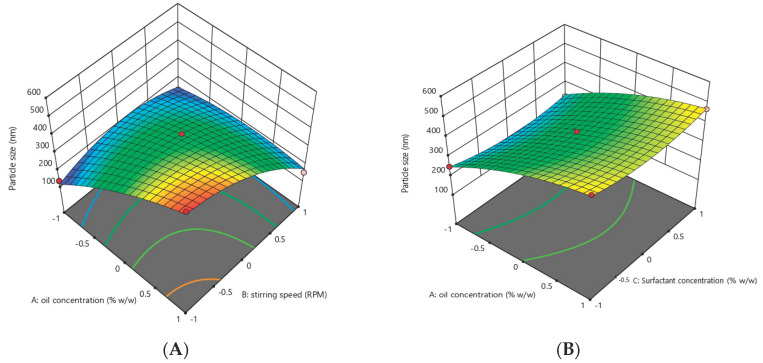
Three-dimensional plot showing effect of independent factors X1 (oil concentration), X2 (stirring speed) and X3 (surfactants concentration) on responses Y1 (particle size) and Y2. (**A**) effect of oil concentration and stirring speed on particle size. (**B**) effect oil concentration and surfactant concentration on particle size. (**C**) effect stirring speed and surfactant concentration on particle size.

**Figure 4 pharmaceutics-14-01640-f004:**
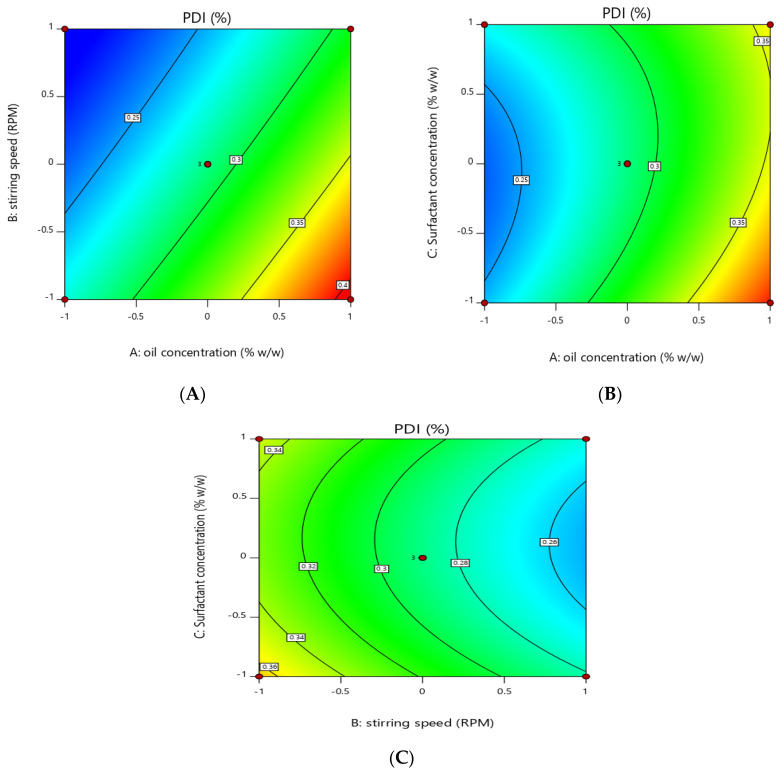
Contour graph showing effect of independent variables on PDI (Y2). (**A**) effect of oil concentration and stirring speed on PDI. (**B**) surfactants concentration and oil concentration on PDI. (**C**) effect of stirring speed and surfactant concentration on PDI.

**Figure 5 pharmaceutics-14-01640-f005:**
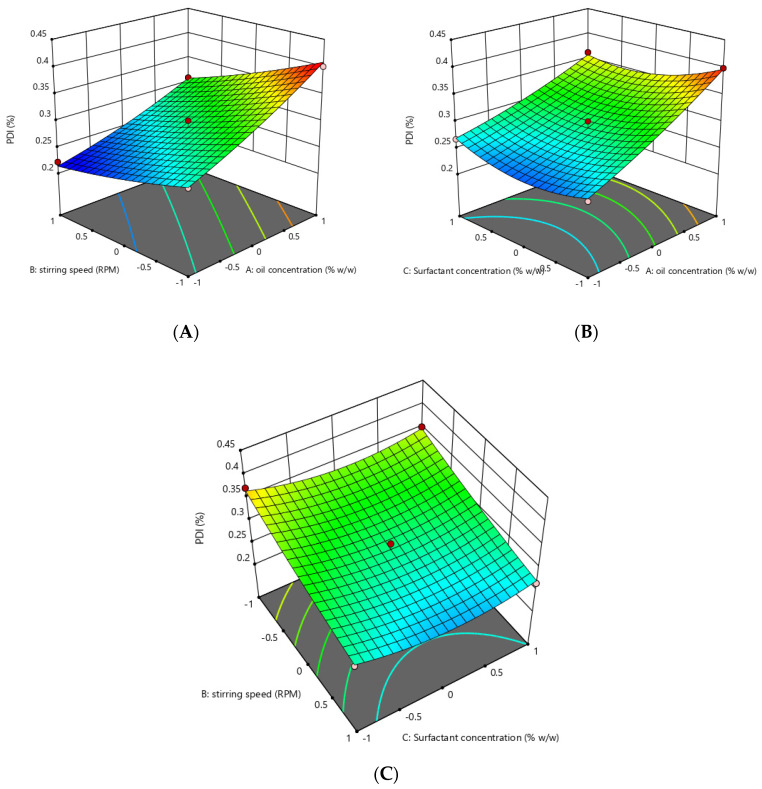
Three-dimensional graph showing effect of independent factors on response PDI (Y2). (**A**) effect of stirring speed and oil concentration on PDI. (**B**) effect of oil concentration and surfactant concentration on PDI. (**C**) effect of stirring speed and surfactant concentration on PDI.

**Figure 6 pharmaceutics-14-01640-f006:**
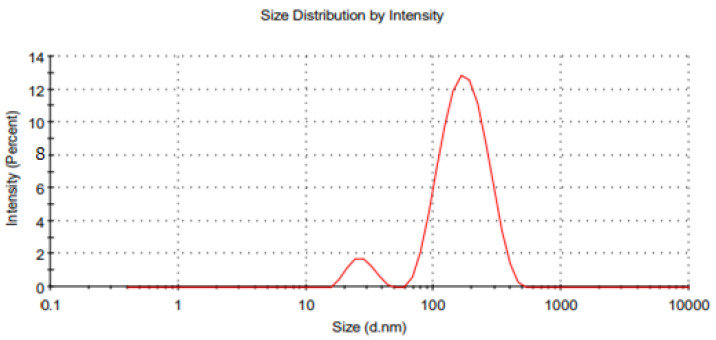
Graph of average particle size distribution of the optimized (F5) essential-oil-loaded nanoemulsion.

**Figure 7 pharmaceutics-14-01640-f007:**
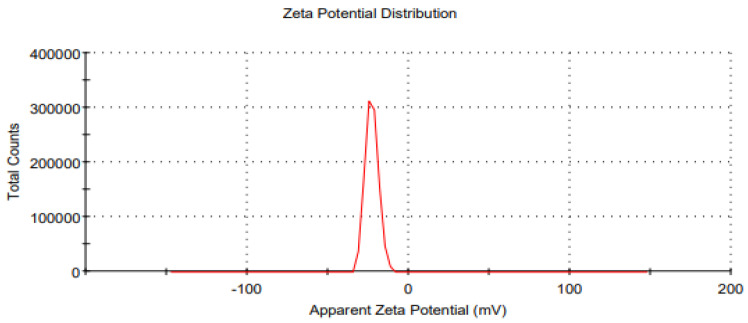
Average zeta potential distribution of the optimized (F5) essential oil loaded nanoemulsion.

**Figure 8 pharmaceutics-14-01640-f008:**
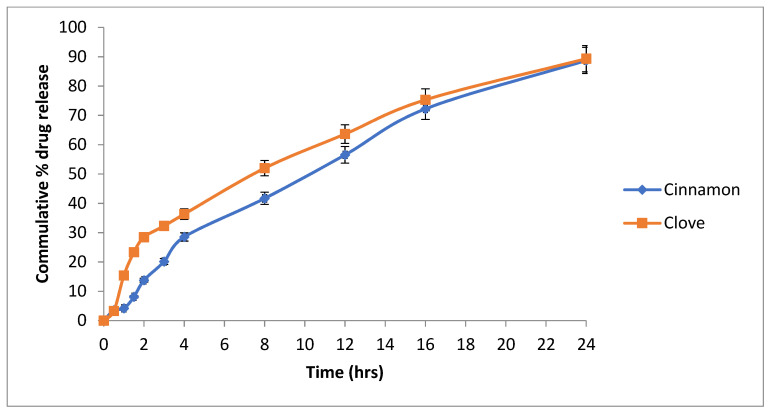
Release profile of the cinnamaldehyde and eugenol from the optimized formulation of nanoemulsion.

**Figure 9 pharmaceutics-14-01640-f009:**
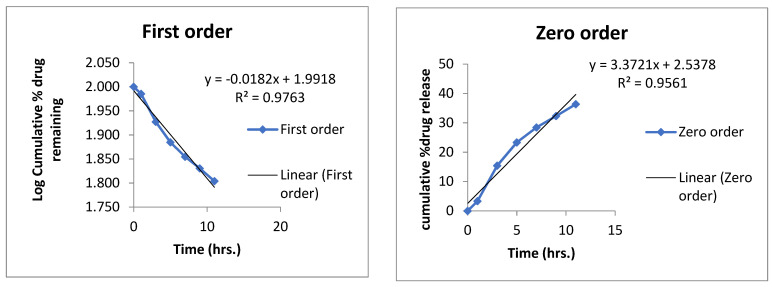
Release mechanism of clove oil from optimized emulsion formulation.

**Figure 10 pharmaceutics-14-01640-f010:**
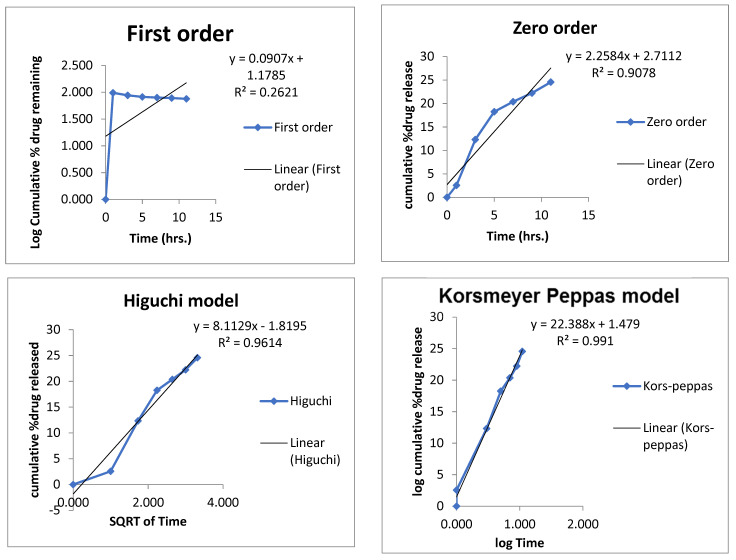
Release mechanism of cinnamon oil from optimized nanoemulsion formulation.

**Figure 11 pharmaceutics-14-01640-f011:**
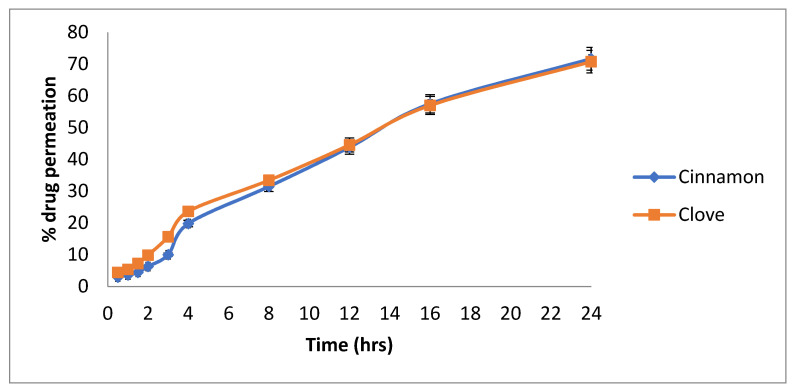
Permeation of cinnamaldehyde and eugenol through a goat buccal mucosa.

**Table 1 pharmaceutics-14-01640-t001:** Independent factors, dependent factors (responses) and parameters.

S.N	Independent Variables (Factors)
**X1**	Oil concentration (% *w*/*w*)
**X2**	Stirring speed (RPM)
**X3**	Surfactantconcentration % *w*/*w*
	**Dependent variables (Responses)**
**Y1**	Globule size
**Y2**	Polydispersity index (PDI)
	**Formulation parameters which were kept constant**
**Z1**	Stirring time 10 min
**Z2**	Essential oil concentration 3 % *w*/*w*
	Level of significance (α) 0.05

**Table 2 pharmaceutics-14-01640-t002:** Independent variables with 3 levels.

Independent Variables	Symbols	Levels		
		−1	0	1
Oil	X1	10	15	20
Stirring speed	X2	10,000	14,000	18,000
Surfactants	X3	1.5	2.5	3.5

−1 = minimum level; 0 = medium level; 1 = maximum level.

**Table 3 pharmaceutics-14-01640-t003:** Different sets of experiments for 15 trails of formulations with coded and actual values.

Run	X1		X2		X3	
	Oil Concentration % *w*/*w*		Stirring Speed RPM		Surfactant Concentration	
	Coded value	Actual value	Coded value	Actual value	Coded value	Actual value
1	0	15	1	18,000	1	3.5
2	1	20	1	18,000	0	2.5
3	1	20	0	14,000	−1	1.5
4	1	20	0	14,000	1	3.5
5	−1	10	1	18,000	0	2.5
6	0	15	−1	14,000	−1	1.5
7	0	15	1	18,000	−1	1.5
8	0	15	0	14,000	0	2.5
9	−1	10	0	14,000	−1	1.5
10	0	15	−1	10,000	1	3.5
11	−1	10	−1	10,000	0	2.5
12	1	20	−1	10,000	0	2.5
13	−1	10	0	14,000	1	3.5
14	0	15	0	14,000	0	2.5
15	0	15	0	14,000	0	2.5

Where 1 for the maximum level, 0 for the medium level and −1 for the minimum level.

**Table 4 pharmaceutics-14-01640-t004:** Design of experiment for Box–Behnken method with 15 trial runs with results of responses.

Run	Factors			Responses	
	X1	X2	X3	Y1	Y2
F1	0	1	1	240	0.266
F2	1	1	0	152	0.309
F3	1	0	−1	481.9	0.399
F4	1	0	1	468.9	0.362
F5	−1	1	0	130	0.222
F6	0	−1	−1	375.7	0.372
F7	0	1	−1	257.6	0.279
F8	0	0	0	393.4	0.288
F9	−1	0	−1	251.4	0.251
F10	0	−1	1	393.4	0.352
F11	−1	−1	0	136.3	0.272
F12	1	−1	0	561.6	0.401
F13	−1	0	1	227.3	0.267
F14	0	0	0	325.1	0.276
F15	0	0	0	351.2	0.301

1 = maximum level, 0 = medium level and −1 = minimum level; X1 (Oil concentration), X2 (Stirring speed), X3 (Surfactants concentration), Y1 (Particle size) and Y2 (Polydispersity index).

**Table 5 pharmaceutics-14-01640-t005:** Build information.

**File Version**	11.0.5.0				
**Study type**	Response surface	**Subtype**	Randomized	**Analysis**	Polynomial
**Design type**	Box–Behnken	**Runs**	15		
**Design model**	**Quadratic**	**Blocks**	No blocks		

**Table 6 pharmaceutics-14-01640-t006:** Summary of 15 runs of formulations.

Response	Name	Units	Minimum	Maximum	Mean	Std. Dev.	Ratio
Y1	P.S	Nm	130	561.6	316.39	131.77	4.32
Y2	PDI	%	0.222	0.401	0.3078	0.0558	1.81

P.S—particle size; PDI—polydispersity index.

**Table 7 pharmaceutics-14-01640-t007:** Regression analysis summary for different models fitting data.

Source	Sequential *p*-Value	Lack of Fit *p*-Value	Adjusted R^2^	Predicted R^2^	
Response Y1					
Linear	0.0047	0.1296	0.5906	0.3406	
2FI	0.0989	0.1796	0.7322	0.3093	
**Quadratic**	**0.0143**	**0.5966**	**0.9397**	**0.7900**	**Suggested**
Response Y2					
Linear	<0.0001	0.3068	0.8832	0.8365	
2FI	0.4096	0.2950	0.8858	0.7915	
**Quadratic**	**0.0373**	**0.6750**	**0.9619**	**0.8810**	**Suggested**

**Table 8 pharmaceutics-14-01640-t008:** ANOVA for Quadratic model of particle size (Y1).

Source	Sum of Squares	df	Mean Square	F-Value	*p*-Value
**Model**	**2.379 × 10^5^**	**9**	**26,429.49**	**25.26**	**0.0012**
X1	1.057 × 10^5^	1	1.057 × 10^5^	100.98	0.0002
X2	59,064.85	1	59,064.85	56.45	0.0007
X3	171.13	1	171.13	0.1635	0.7026
X1X2	40,662.72	1	40,662.72	38.86	0.0016
X1X3	30.80	1	30.80	0.0294	0.8705
X2X3	311.52	1	311.52	0.2977	0.6088
X1^2^	4639.04	1	4639.04	4.43	0.0891
X2^2^	21,408.69	1	21,408.69	20.46	0.0063
X3^2^	4853.04	1	4853.04	4.64	0.0839
**Residual**	5231.93	5	1046.39		
**Cor Total**	2.431 × 10^5^	14			

df—degree of freedom.

**Table 9 pharmaceutics-14-01640-t009:** ANOVA for Quadratic model of PDI (Y2).

Source	Sum of Squares	Df	Mean Square	F-Value	*p*-Value
**Model**	**0.0430**	**9**	**0.0048**	**40.28**	**0.0004**
X1	0.0263	1	0.0263	222.08	<0.0001
X2	0.0129	1	0.0129	108.62	0.0001
X3	0.0004	1	0.0004	3.07	0.1399
X1X2	0.0004	1	0.0004	3.72	0.1117
X1X3	0.0007	1	0.0007	5.92	0.0591
X2X3	0.0000	1	0.0000	0.1033	0.7609
X1^2^	0.0002	1	0.0002	1.79	0.2385
X2^2^	0.0001	1	0.0001	0.8046	0.4108
X3^2^	0.0021	1	0.0021	17.69	0.0084
**Residual**	0.0006	5	0.0001		
**Cor Total**	0.0436	14			

**Table 10 pharmaceutics-14-01640-t010:** Zeta potential of the formulations.

F/Code	Zeta Potential (mV)	F/Code	Zeta Potential (mV)
F1	−15.51	F9	−14.76
F2	−19.54	F10	−8.7
F3	−7.87	F11	−21.33
F4	−7	F12	−5.72
F5	−22.9	F13	−17
F6	−9.5	F14	−12.75
F7	−14	F15	−11
F8	−8.32		

**Table 11 pharmaceutics-14-01640-t011:** pH of the 15 formulations of the nanoemulsion.

F/Code	pH	F/Code	pH
F1	6.8	F9	6.6
F2	6.7	F10	6.6
F3	7.2	F11	6.8
F4	7.3	F12	6.9
F5	6.4	F13	7.1
F6	6.2	F14	7.2
F7	6.9	F15	7.2
F8	7		

**Table 12 pharmaceutics-14-01640-t012:** Kinetic stability of nanoemulsion.

Stirring Speed *	Phase Separation	Creaming	Flocculation
1000	Not detected	Not detected	Not detected
2000	Not detected	Not detected	Not detected
3000	Not detected	Not detected	Not detected

* revolutions per minute.

**Table 13 pharmaceutics-14-01640-t013:** Encapsulation efficiency of optimized nanoemulsion.

Formulation	Active Ingredient	Eneterapmentefficiency
Nanoemulsion	Cinnamon oil	97.43%
Nanoemulsion	Clove oil	98%

**Table 14 pharmaceutics-14-01640-t014:** Kinetic models application on formulated nanoemulsion.

Formulation	Active	Zero	First	Higuchi Model	Korsmeyer Peppas Model		Best Fitt Model
		R^2^	R^2^	R^2^	R^2^	n	
Nanoemulsion	Cinn	0.9078	0.2621	0.9614	0.991	0.734	Korsmeyer–Peppas model
	Eug	0.9561	0.9763	0.9662	0.9919	0.522	

Cinn—cinnamon; Eug—eugenol.

**Table 15 pharmaceutics-14-01640-t015:** Antibacterial activity and Antiquorum sensing of essential-oil-loaded nanoemulsion.

Strain	Formulation	Zone of Inhibition (mm)
*Staphylococcos epidermidis*	Loaded nanoemulsion	9
*Staphylococcos epidermidis*	Un loaded nanoemulsion	0
*Staphylococcos aureus*	Loaded nanoemulsion	9
*Staphylococcos aureus*	Un loaded nanoemulsion	0
*Chromobacerium voilaceum*	Loaded nanoemulsion	18
*Chromobacerium voilaceum*	Un Loaded nanoemulsion	0

Ciprofloxacin 14 mm against *Staphylococcos epidermidis* and 16 mm against *Staphylococcos aureus* and 16 mm against *Chromobacerium voilaceum*.

**Table 16 pharmaceutics-14-01640-t016:** Skin irritation study of the optimized nanoemulsion formulation.

Rats	Time 0 h			Time 24 h			Time 48 h		
	I	R	E	I	R	E	I	R	E
Nanoemulsion	N	N	N	N	N	N	N	N	N
Formalin	N	N	N	Y	Y	Y	Y	Y	Y
Negative control *	N	N	N	N	N	N	N	N	Ns

I—skin irritation; R—skin redness; E—skin erythema; N—absent; Y—present; * blank formulation.

## Data Availability

Not applicable.

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
