# Peer review of "Fabrication and Optimization of Essential-Oil-Loaded Nanoemulsion Using Box–Behnken Design against Staphylococos aureus and Staphylococos epidermidis Isolated from Oral Cavity"

_pharmaceutics, 2022, doi:10.3390/pharmaceutics14081640_

Round 1
Reviewer 1 Report
Dear Authors,
I suggest it is still good for authors to review their manuscripts.
1. I asked, why in quadratic equations, some variables were insignificant? What had influenced that? Your answer: All variables were added to see the effect on responses that are essential for formulation design, and can’t be omitted out. It is important to note that in this case individual and combined effects are included in the results.
2. What allows you to say that the evaporation of essential oils is negligible from nanoemulsions? Your answer: Since essential oils were initially encapsulated in the olive oil and then further process was accomplished. Thus evaporation was negligible.
3. I suggested using oil particles instead of oil droplets. However, this was not changed everywhere in the text.
Author Response
Reviewer 1
Comment |
Remarks
|
Reviewer 1
1.I asked, why in quadratic equations, some variables were insignificant? What had influenced that? Your answer: All variables were added to see the effect on responses that are essential for formulation design, and can’t be omitted out. It is important to note that in this case individual and combined effects are included in the results.
2. What allows you to say that the evaporation of essential oils is negligible from nanoemulsions? Your answer: Since essential oils were initially encapsulated in the olive oil and then further process was accomplished. Thus evaporation was negligible.
3. I suggested using oil particles instead of oil droplets. However, this was not changed everywhere in the text. |
1.This is software generated quadratic equation. The software compares various variables besides considering individual variables. Since our p-value (<0.05) was considered as level of significance, so all individual variables were significant. whereas when variable were combined by software the results were insignificant it uniqueness of design expert that it compares individual and combined variables variable by itself. The data revealed individual variable significance in manuscript we proceeded with individual variable and calculated response.
Above explanation is also added in the text.
2. Since essential oils were initially encapsulated in the olive oil and then further process was accomplished. Thus evaporation was negligible. Further The nano emulsion formulation was executed by placing the aluminium foil over beakers that prevented the evaporation.
We have changed it already in manuscript.
|

Reviewer 2 Report
In this paper, response surface methodology (RSM) was used to design and optimize the formulation of nano-emulsion containing essential oil, and the optimal formulation of nano-emulsion was investigated, mainly exploring its active ingredients, particle size, zeta potential, stability, antibacterial performance and skin irritation. There are many errors in the paper, such as lack of data, text description inconsistent with the experimental results, and many improper formats, so it is suggested to reject the paper.
1. The reasons for setting oral pH in 2.7.6 are lack of literature support, and the pH range in 3.6 results is inconsistent with that in the table, with pH values of 6.2 and 6.4 appearing in the table.
2. In 2.7.9 experiments related to drug release, phosphoric acid buffer with pH 6.8 was mentioned to simulate oral environment. However, oral cavity has unique anatomical structure, and different parts have different pH, oxygen content, humidity and other microenvironments, so this simulation cannot be called oral physiological environment simulation.
3. In 2.7.11 skin irritation test, the optimized formula was used for the experiment, formalin was used as the positive control, but there was no negative control in the experiment, and the specific site of treatment was not explained. The epidermis in the oral cavity was different from the epidermis in the body.
4. In 2.9 why did Staphylococcus superficial and Staphylococcus aureus be changed to C. violaceum in the anti-quorum sensing experiment, and corresponding experimental date did not appear in the follow-up results.
5. There is a lack of date in 3.7 of the result. The text describes the appearance of emulsions with two different configurations of F5 and F4, but there are only two graphs corresponding to the data, and the data of which kind of emulsion is not marked.
6. The contents of the table in 3.17 results are inconsistent with the results of the supplementary plate bacteriostatic zone and lack of experimental groups, which makes it difficult to prove the significant bacteriostatic effect mentioned in the results.
7. There are many formatting errors in the text, such as -1 in cm-1 in 2.4 is not superscripted; all tables in the text should be three-line table; in 3.1, there should be no scale on the ordinate of the FTIR graph; in Fig 11, the scale is missing and most of the pictures are blurry, please upload high-resolution pictures and modify them carefully.
Author Response
Reviewer 2
Comment |
Remarks
|
1. The reasons for setting oral pH in 2.7.6 are lack of literature support, and the pH range in 3.6 results is inconsistent with that in the table, with pH values of 6.2 and 6.4 appearing in the table. 2. 3. 4. In 2.7.9 experiments related to drug release, phosphoric acid buffer with pH 6.8 was mentioned to simulate oral environment. However, oral cavity has unique anatomical structure, and different parts have different pH, oxygen content, humidity and other microenvironments, so this simulation cannot be called oral physiological environment simulation. 5. 6. In 2.7.11 skin irritation test, the optimized formula was used for the experiment, formalin was used as the positive control, but there was no negative control in the experiment, and the specific site of treatment was not explained. The epidermis in the oral cavity was different from the epidermis in the body. 7. 8. 9. 10. 11. 12. 13. 14. 15. In 2.9 why did Staphylococcus superficial and Staphylococcus aureus be changed to C. violaceum in the anti-quorum sensing experiment, and corresponding experimental date did not appear in the follow-up results.
16.
17. There is a lack of date in 3.7 of the result. The text describes the appearance of emulsions with two different configurations of F5 and F4, but there are only two graphs corresponding to the data, and the data of which kind of emulsion is not marked. 18. 19. 20. The contents of the table in 3.17 results are inconsistent with the results of the supplementary plate bacteriostatic zone and lack of experimental groups, which makes it difficult to prove the significant bacteriostatic effect mentioned in the results.
21. There are many formatting errors in the text, such as -1 in cm-1 in 2.4 is not superscripted; all tables in the text should be three-line table; in 3.1, there should be no scale on the ordinate of the FTIR graph; in Fig 11, the scale is missing and most of the pictures are blurry, please upload high-resolution pictures and modify them carefully. |
We agree with reviewer’s comments and Relevant reference (Oral cavity) is cited as [30] [Relevant reference is added], further the section 3.6 is revised removing mistakes.
The application site of our formulation is oral cavity (buccal mucosa). To mimic buccal mucosal pH (6.8), we performed this experiment specifically on 6.8, that is also cites as reference [30].
Irritation test employed is popular to be called as skin irritation test (Draize test) for topical application. This test was performed to check any sort of irritation on skin in general if the same formulation would apply for skin and soft tissue infections due to antimicrobial nature of our formulation.. However, skin (buccal mucosa ) irritation potential of the optimized formulation was investigated to note compatibility of formulation in oral mucosa of goat oral cavity. Further The blank formulation was used as negative control. The C. violaceum is an indicator strain of bacterial Quorum sensing. Literature can be checked that while determining anti QS activity of strains, generally C.violaceum strain is used. Further, since our both clinical strains are Biofilm producers, an inhibition of C-violaceum gives an idea regarding antibiofilm activities too, that is further confirmed by antibiofilm studies of both clinical tested strains. In conclusion, use of C.violaceum strain was only performed to check cell –cell signaling as obvious from literature.
Formulation F5 is specified since it is optimized formulation. This point is highlighted in the text.
We agree with reviewer’s comments since it’s a mistake. Data of Table 16 is more clarified by adding 2 rows in respective bacterial data, and data is now consistent with images.
Manuscript is revised once again all mistakes are removed and highlighted. All table are corrected as three line. FTIR ordinate scale is removed. Images are added with more clarity
|

Reviewer 3 Report
The discussion about oral infections is feeble. I strongly recommend that authors read specific manuscripts about it. One good example is Mira A, Simon-Soro A, Curtis MA. Role of microbial communities in the pathogenesis of periodontal diseases and caries. J Clin Periodontol. 2017 Mar;44 Suppl 18:S23-S38. doi: 10.1111/jcpe.12671. PMID: 28266108.
Author Response
Reviewer 3
Comment |
Remarks
|
The discussion about oral infections is feeble. I strongly recommend that authors read specific manuscripts about it. One good example is Mira A, Simon-Soro A, Curtis MA. Role of microbial communities in the pathogenesis of periodontal diseases and caries. J Clin Periodontol. 2017 Mar;44 Suppl 18:S23-S38. doi: 10.1111/jcpe.12671. PMID: 28266108. |
Indeed this is worth mentioning article and we have read it very carefully. Based on importance of article, we have cited this article in manuscript as [60] |

Round 2
Reviewer 2 Report
In this paper, response surface methodology (RSM) was used to design and optimize the formulation of nano-emulsion containing essential oil, and the optimal formulation of nano-emulsion was investigated, mainly exploring its active ingredients, particle size, zeta potential, stability, antibacterial performance and skin irritation. The author has revised the manuscript and responded the reviewer’s comments and supplemented the associated data. The revised article has reached the publication level and can be accepted.
This manuscript is a resubmission of an earlier submission. The following is a list of the peer review reports and author responses from that submission.
Round 1
Reviewer 1 Report
The manuscript by Niamat Ullah et al. deals with "Design, fabrication, and optimization of essential oil loaded nanoemulsion using Box-Behnken design for oral bacterial infection".
The conclusion of this studies reveals the truth. There was basically no conclusion in this manuscript (. Conclusions : It was concluded that cinnamon and clove essential oil based formulation showed promising antibiofilm activity infections through encapsulation and delayed-release from the olive oil using olive oil water nanoemulsion.)
I would to like to see a great conclusive remarks from authors side to make this publication acceptable.
Regarding the FTIR studies, please explain the main peaks ( which functional groups). Please include the structure of the main component or compound from each essential oils.
Reviewer 2 Report
This study aimed to formulate a stable essential oil-loaded nanoemulsion for treatment of oral bacterial infections. The authors performed several tests to physic-chemically characterize their formulations. The authors also performed two antimicrobial test: zone of inhibition and anti-quorum sensing. As a result, the particle size, polydispersity index, and zeta potential of 33 the optimized formulation were 130mm, 0.222, and -22.9 respectively. The ATR-FTIR 34 analysis revealed that there was no incompatibility between the active ingredients and 35 the excipients. A significant release profile in active ingredients of nanoemulsion i.e 88.75 36 % of the cinnamaldehyde and 89.33 % of eugenol was recorded after 24hrs. In the ex-vivo 37 goat mucosal permeation study, 71.67 % of the cinnamaldehyde permeated and 70.75% 38 of the eugenol permeated from the nanoemulsion.
The antimicrobial tests showed a 9 mm zone of inhibition against Staphylococcus aureus and Staphylococcus epidermidis. Whereas in anti-quorum sensing analysis the optimized nanoemulsion formulation showed an 18 mm zone of inhibition. The authors concluded this project confirmed that essential oil loaded nanoemulsion produced promising results against tested oral bacteria and thus can be a potential therapy against oral infections.
The antimicrobial tests conducted here and respective results do not support the conclusion. It is inadequate to achieve the objective because the bacteria evaluated were not the main ones associated with the most prevalent oral diseases such as caries and periodontal disease. This reviewer suggests modifying the title, the objective and the conclusion by removing oral infections from them. Another option is to keek these parts but includes P. gingivalis and S. mutans as the bacterial species evaluated in the methodology since these bugs are associated with oral diseases and systemic diseases.
Besides, the zone of inhibition is a preliminary test. As the authors mentioned in the introduction, "Biofilm forming bacterial communities are more dangerous as compared to planktonic bacteria as these are more resistant to the external stress due to the biofilm protecting coat". Therefore, a multispecies biofilm assay is necessary to increase the scientific and clinical relevance of the present manuscript. The discussion must include the tested bacteria species' role in oral infections and compare the current antimicrobial results with other results from the literature.
At last, the antimicrobial tests were not described in the Material and methods section of the present manuscript.
Reviewer 3 Report
1. In introduction, there is no description why and the significance of nanoemulsion was used.
2. From line 92 to 96, what is the strain source of Staphylococcus epidermidis? Why Pseudomonas aeruginosa was shown here? Pseudomonas aeruginosa was not found in this work.
3. In title, this work was aimed against oral infections, but Staphylococcus epidermidis and Chromobacterium violaceum are not found in oral. Why these strains are chosen?
4. How to prepare the essential oil loaded nanoemulsion? Authors showed many characterizations of nanoemulsion, but essential oil loaded nanoemulsion was few.
5. In 2.7.5, what the meaning of title “centrifugation”? “centrifugation” is the property of nanoemulsion? This section showed the macroscopic stability.
6. In Fig 1, “Abs” means? Wavelength (λ)and wave number (σ)were used in many reports. And what is the “LNE”, “ULNE”? authors should give the full name.
7. In Fig 6 and Fig 7, authors should give the charts by graph software and provide the “Data Analysis”.
8. In Table 16, there was antiquorum sensing activity essential oil loaded emulsion but no emulgel?
9. In Fig12, how to ensure that the diameter of unloaded nanoemulsion or essential oil loaded nanoemulsion is consistent? The diameter is?
10. In Fig 13 and 14, the culture conditions S. aureus and S. epidermidis is consistent? Why there were obvious colonies in Fig 13A, but not in Fig 13B, Fig 14A and 14B. what is the amount of S. aureus and S. epidermidis used to plate spread?
11. What is the novelty in this work? Because the preparation of nano-emulsion was referred by pervious method, and authors do not show the specific advantages of this essential oil loaded nanoemulsion.
12. In Table 12, what is “Stirring rpm”?
13. Some minor suggestions need to correct. For example, bacterial names should be in italics (line 58, 60, etc.,); “Chromobaceriumvoilaceum” in line 496; consistency of statements (“RPM” and “rpm” in Table 1 and 12; “time” and “Time” in Fig 9 and 10).
Reviewer 4 Report
Dear Authors,
I found this manuscript interesting, but I have several comments:
- I propose to change the term water to purified water according to the European Pharmacopoeia. Line 105
- Remove the Y2 at the end of the sentence ... was investigated. Line 121
- Where Tables 1 and 3 are given in the text?
- Why were essential oils not evaluated by gas chromatography? Lines 167–172
- At what time intervals were the samples taken? Lines 186 and 202
- At 285 or 290 nm for a cinnamon essential oil? Line 203
- Has a statistical analysis been performed? If yes, then I suggest including it in 2. Materials and Methods.
- Where Figure 1 is given in the text? A detailed explanation of this Figure 1 would be needed.
- Maybe it should be Table 4 instead of Table 6. Lines 239–242
- Why were statistically insignificant variables (e.g., surfactants concentration) included in the quadratic equations? Lines 305–306, 362–363
- Why this section 3.5 Zeta potential of the formulation is not described, only the table is given? Line 370
- Where Table 11 is given in the text?
- What is the composition of the optimized nanoemulsion? Line 385
- Where Figures 6 and 7 are given in the text? A detailed explanation of these Figures would be needed.
- How did you assess whether the essential oil (it is volatile) did not evaporate from the nanoemulsion?
- Where Figures 12, 13, and 14 are given in the text? A detailed explanation of these Figures would be needed.
- Where Table 17 is given in the text?
- I suggest using oil particles instead of oil droplets.
- The Discussion presents some of the results of the study, which should be moved to the Results section.
Reviewer 5 Report
In my opinon, the are covered in this paper is interesting, and suitable for this journal. However, the manuscript needs to be revised in deep. there are some typos that make difficults its reading. The introduction is too general and not specific for the manuscript, I mean explain low level things and no mentioning other points important for the topic. There are many figures and tables taht repeat information. Here there are a couple of comments that can help
Minor points:
- Please, avoid repeating words in close sentences (e.g abstract therefore) . In this case, the second one should be deleted. Line 38 with permeated…
- Line 42: I suggest removing the sentence “The result of this project”
- Line 53: worldwide must nor be separated.
- Line 56: Please, include space between words. Same line 178
Major points:
- Introduction: The authors must explain why the are using the specific essential oil as antibiotics and why the development of nanoemulsions. The introduction is too general, low level and not adequate for this manuscript. Must be re-written.
- The elaboration of nanoemulgel must detailed in methods. Did the authors prepared nanoemulsions and nanoemulgel?
- Line 187: Please, define physiological circumstances. Oral cavity?
- A section with statistical analysis should be included.
- In my opinion, manuscripts should be impersonal. Eg. Line 216.
- Line 265: “on the particle size of the nanoemulsion means 265 particle size” This sentence must be re-written.
- Korspeppas model I guess the authors mean Korsmeyer Peppas Model?
- The manuscript is full of tables and figures. The point is that many of them does not provide “additional” information. Eg. Fig 9 and 10. , Fig12-14 ( should be supplemenatary). Tables 16-18 can be combined…